# Hydrogel Films Based on Chitosan and Oxidized Carboxymethylcellulose Optimized for the Controlled Release of Curcumin with Applications in Treating Dermatological Conditions

**DOI:** 10.3390/molecules26082185

**Published:** 2021-04-10

**Authors:** Mohamed Dellali, Camelia Elena Iurciuc (Tincu), Corina Lenuța Savin, Nawel Spahis, M’hamed Djennad, Marcel Popa

**Affiliations:** 1Laboratory of Structure, Elaboration, and Application of Molecular Materials, Abdelhamid Ibn Badis University of Mostaganem, Mostaganem 27000, Algeria; m.dellali@univ-chlef.dz (M.D.); mhamed.djennad@univ-mosta.dz (M.D.); 2Faculty of Technology, Hassiba Benbouali University of Chlef, BP 151, Chlef 02000, Algeria; n.spahis@univ-chlef.dz; 3Department of Natural and Synthetic Polymers, Gheorghe Asachi Technical University of Iasi, Mangeron Blvd. no. 73, 700050 Iasi, Romania; savincorina@yahoo.com; 4Department of Pharmaceutical Technology, Faculty of Pharmacy, “Grigore T. Popa” University of Medicine and Pharmacy, University Street, no. 16, 700115 Iaşi, Romania; 5Academy of Romanian Scientists, Splaiul Independentei Street, No 54, 050094 Bucharest, Romania

**Keywords:** chitosan, oxidized carboxymethyl cellulose, hydrogel film, curcumin immobilization, delivery system

## Abstract

Cross-linked chitosan (CS) films with aldehyde groups obtained by oxidation of carboxymethyl cellulose (CMC) with NaIO_4_ were prepared using different molar ratios between the CHO groups from oxidized carboxymethyl cellulose (CMCOx) and NH_2_ groups from CS (from 0.25:1 to 2:1). Fourier-transform infrared (FTIR) and nuclear magnetic resonance (NMR) spectroscopy demonstrated the aldehyde groups’ presence in the CMCOx. The maximum oxidation degree was 22.9%. In the hydrogel, the amino groups’ conversion index value increased when the -CHO/-NH_2_ molar ratio, cross-linking temperature, and time increased, while the swelling degree values decreased. The hydrogel films were characterized by scanning electron microscopy (SEM) and FTIR analysis. The curcumin encapsulation efficiency decreases from 56.74% to 16.88% when the cross-linking degree increases. The immobilized curcumin release efficiency (R_Ef_%) and skin membrane permeability were evaluated *in vitro* in two different pH solutions using a Franz diffusion cell, and it was found to decrease when the molar ratio -CH=O/NH_2_ increases. The curcumin R_Ef_% in the receptor compartment was higher at pH = 7.4 (18%- for the sample with a molar ratio of 0.25:1) than at pH = 5.5 (16.5%). The curcumin absorption in the skin membrane at pH = 5.5 (47%) was more intense than at pH = 7.4 (8.6%). The curcumin-loaded films’ antioxidant activity was improved due to the CS presence.

## 1. Introduction

Dermatology is a constantly evolving medical field that deals with skin, nail, hair diseases, and systemic diseases treatment, especially for those whose symptoms can be observed primarily on the skin. The treatments performed aim to improve the skin’s external appearance by treating various diseases and pathological conditions. Although skin disease incidence is lower than other diseases, they significantly impact life quality, and skin cancer or severe infections are life-threatening [1]. The most common skin diseases reported during a patient’s lifetime were warts (41.3%), acne (19.2%), and contact dermatitis (15.0%), followed by other forms of eczema or atopic dermatitis and urticaria [2].

Hydrogels are three-dimensional polymeric structures that can absorb large amounts of water or biological fluids. One of the main advantages of the hydrogels used in the local treatment of skin diseases is their easy applicability and the possibility of reducing the immobilized drugs’ side effects. During topical (dermal) administration, the drug initially penetrates the stratum corneum, then diffuses into the deeper epidermis until it reaches the dermis, being absorbed into the systemic circulation through dermal microcirculation [3].

The drug’s local administration depends on many factors, including the skin’s barrier properties, the incorporated therapeutic compound’s physicochemical properties, and the delivery system used [4]. The immobilized drug interaction with the polymer matrix is essential for drug effectiveness in topical applications. The application of these delivery systems with immobilized drugs results in the therapeutic concentration achievement in the individual layers of the skin located in the application area. At the same time, the serum concentration is low, which causes a decrease in the drug’s side effects [5,6].

Natural polymer-based (proteins and polysaccharides) hydrogels are most often used in biomedical applications due to their biocompatibility, non-toxicity, biodegradability, the similarities of their physical properties to natural tissues, low immunogenicity, and their functional groups could be modified to obtain release systems with improved properties [1].

Polysaccharides offer a wide range of versatile functionalities and structural diversity due to their variable molecular weight and numerous and varied functional groups (amino, carboxyl, carbonyl, and hydroxyl) on the main chain [7]. These hydrophilic groups determine an increased solubility in water and strengthen their bioadhesive and biorecognition characteristics (electrostatic interactions between biological tissues and polysaccharides) [8]. For example, chitosan, the only polysaccharide with a positive charge (polycation), can be attached to negatively charged mucosal layers through electrostatic interactions [9,10].

Chitosan (CS) is a natural, cationic polysaccharide obtained by chitin’s alkaline deacetylation process [11]. It is a semicrystalline, semisynthetic polymer with a linear structure, composed of (1→4)-2-acetamido-2-deoxy-β-D-glucan (N-acetyl-D-glucosamine) and (1→4)-2-amino-2-deoxy-β-D-glucan (D-glucosamine) units [12]. It has excellent properties such as biodegradability, biocompatibility, non-toxicity, and mucoadhesivity [13,14]. The CS has high reactivity and can be easily functionalized due to the hydroxyl and amine groups found in large numbers along its chain and allow the reaction with different cross-linking agents. Also, chitosan itself possesses biological activity. The hydrogels obtained are used as delivery systems for the controlled release of drugs and genes [15,16], in biosensors obtaining [17], and in medical imaging [18,19].

The hydrogels’ properties and the release of the drug from these delivery systems depend on the type of cross-linking agent used and its concentration. The morphology of such hydrogels is porous and depends on the cross-linking agent used [20].

Aldehydes, epoxy compounds, esters are chemical cross-linking agents commonly used to prepare CS-based hydrogels, and sodium tripolyphosphate or sodium sulfate have been used as ionic cross-linking agents [21,22,23]. In an aqueous solution, the aldehydes form an imine bond with the amine group in the CS to give a biocompatible hydrogel. Glutaric aldehyde acts as a cross-linking agent, and it has been shown that the morphology of the obtained hydrogel is porous, and the pores are evenly distributed in the formed matrix [24]. Glutaric aldehyde often causes neurological and other cytotoxicities [25]. Macromolecules containing aldehydes can be used to prepare CS hydrogels [26]. Thus, poly(ethylene glycol) dialdehyde was used in the CS hydrogel and has been shown to improve the polymer matrix mechanical properties as the cross-linking agent’s concentration increases [27,28]. Natural carbonyl compounds, including partially oxidized polysaccharides, are a preferred alternative as cross-linking agents, thus increasing the application area for the hydrogels obtained [29]. Research has shown that amino groups in CS react with aldehyde groups in oxidized cellulose to form Schiff bases [30].

This paper reports the results obtained after preparing a biocompatible hydrogel based on CS cross-linked with oxidized carboxymethyl cellulose (CMCOx) in which curcumin was incorporated. Previous in vivo research and clinical trials have shown that CS-based hydrogel films with topical applications can cause rapid wound healing, stop bleeding, reduce healing time, faster vascularization, and colonization with fibroblasts were observed [31]. Curcumin was chosen as an active principle model used to treat skin diseases due to its remarkable therapeutic properties and strong antioxidant character [32,33]. The main disadvantage of curcumin is its insolubility in water and low bioavailability in cells. Curcumin can degrade rapidly under the influence of several factors such as natural light, alkaline pH. It is a good metal chelator, but previous studies have shown that curcumin-based metal complexes cause a decrease in their pharmacological action compared with free curcumin [34]. Curcumin can be protected from degrading factors by its encapsulation in various pharmaceutical formulations such as release systems based on natural and synthetic polymers, liposomes, micro/nanoemulsions, solid lipid nanoparticles [35]. Lipids associated with curcumin indeed increase bioavailability, but particles of this kind do not ensure a continuous, sustained release, even if the amount of immobilized curcumin is higher compared with biopolymers nanoparticles [36,37]. We choose CS to obtain the hydrogel because it has antimicrobial activity and antioxidant activity, which is an advantage over other polymers, including polysaccharides.

Oxidized derivatives (CMCOx) were obtained by converting 1,2-dihydroxyl groups into dialdehyde groups at C2 and C3 without significant side reactions under the action of NaIO_4_ [38]. Fourier-transform infrared (FTIR) and nuclear magnetic resonance (NMR) spectroscopy identified the modified polysaccharide structure. The reverse titration reaction with Na_2_S_2_O_3_ was used to determine the aldehyde groups’ content and the polysaccharide’s oxidation degree. The CMCOx molecular weight was determined by the viscosimetric method. Several hydrogels were obtained by varying the molar ratio between the -NH_2_ groups from CS and -CHO groups from CMCOx.

In order to obtain the hydrogels, the reaction temperature, the molar ratio between CS and CMCOx (practically, between the number of moles of -CH=O and -NH_2_ groups), and the cross-linking time were the parameters whose influence was studied. The different classical characterization methods make it possible to define between the hydrogel films obtained, those with the best water retention capacity, and the best porosity for the active principle’s encapsulation and controlled release. The structural characterization and study of hydrogel films’ morphology are detailed in this article.

This article’s main objective and point of originality were to optimize the reaction conditions to obtain CS-based hydrogel films cross-linked with CMCOx with the best physicochemical characteristics and use them as delivery systems for controlled and sustained release of curcumin. The hydrogels obtained are sensitive to pH and have the ability to encapsulate and release the active principle. The release kinetics and permeability of curcumin from the obtained hydrogels were studied on chicken skin used as a membrane in two different pH environments, at pH = 5.5 and pH = 7.4 at 37 °C up to equilibrium using a Franz diffusion cell. The antioxidant activity, expressed by IC50 (the inhibition concentration at which the DPPH radicals were scavenged by 50%) calculated from the interpolation of the linear regression analysis for curcumin and hydrogel films with or without encapsulated curcumin was determined. The influencing factors leading to the intensification of antioxidant activity in hydrogel films with immobilized curcumin were studied.

## 2. Results and Discussions

### 2.1. Preparation and Characterization of CMCOx

Sodium periodate has been used to oxidize the C2-C3 vicinal hydroxyl groups in the carboxymethyl cellulose (CMC) glucoside cycle, but glycosidic bonds’ cleavage is also inevitable leading to polymer degradation [39,40]. The oxidation reaction did not form toxic compounds [41], and it took place in the dark to avoid the advanced oxidation, in double-distilled water pH = 6.5 and a temperature of 30 °C. Appendix A schematically shows CMC’s oxidation reaction.

#### 2.1.1. FTIR Spectroscopy of CMC and CMCOx

FTIR spectra were recorded to demonstrate CMC oxidation by the presence of aldehyde groups in the modified polysaccharide. Figure 1 shows the FTIR spectra of CMC and CMCOx for 6 h.

The absorption peak from 1599 cm^−1^ in the CMC spectrum appears shifted in the CMCOx spectra to 1631 cm^−1^. Absorption at 1599 cm^−1^ can be assigned to carboxylate anion (νc = o) asymmetric stretch band from the glucuronic acid in the CMC spectrum [42], and the absorption bands’ appearance at a lower frequency from 1631 cm^−1^, which is characteristic for conjugated C=O, from carboxylate anion and could be determined by CMCOx water absorption and degradation [43]. The appearance of a new absorption peak of a lower intensity from 1677 cm^−1^ attributed to unsaturated ketones groups is a clear indication that partial oxidation of CMC has occurred [44,45]. The absorption peak from 1725–1740 cm^−1^, specific to the aldehyde group [44], did not occur because there is the possibility that aldehydes groups were found in the hemiacetals forms in CMCOx; the appearance of the peak could be masking it. For example, in the case of dextran and inulin oxidation, the authors could not identify in the FTIR spectrum the absorption band specific to the aldehyde group [46,47]. Absorption bands from about 800 cm^−1^ can be attributed to semiacetals forming between aldehyde groups and vicinal hydroxyl groups [48,49]. The absorption band from 3593 cm^−1^ correspondingly to the OH-group’s elongation vibration in the CMC spectrum becomes narrower in the CMCOx spectra. The absorption band moves to approximately 3472 cm^−1^, which probably indicates that the hydroxyl groups number has decreased due to the aldehyde group formation after the oxidation reaction. The absorption bands from 1424 cm^−1^ to 1328 cm^−1,^ also attributed to the carboxylate anion in the CMC spectrum, are found in the CMCOx spectrum but appear slightly shifted, indicating structural changes following the oxidation reaction.

#### 2.1.2. NMR Spectra of CMC and CMCOx

^1^H NMR and ^13^C NMR spectra were recorded for both CMC and CMCOx. The spectra are shown in Figure 2 (^1^H NMR) and Figure 3 (^13^C NMR). 

^1^H NMR and ^13^C NMR spectroscopy were used to assess whether the CMC oxidation had occurred and identified if the signals corresponding to the aldehyde groups were found in the CMCOx spectra. Figure 2 shows that the ^1^H NMR spectra of the CMCOx sample (the oxidation reaction time for this sample was 6 h) dissolved in deuterated water or DMSO present characteristic peaks similar to those in the CMC spectrum. The formation of aldehyde groups in the structural unit of CMC is highlighted in the ^1^H NMR spectrum of CMCOx dissolved in water by the presence of the singlet signal at 9.24 ppm and in the CMCOx spectrum dissolved in DMSO at 10.11 ppm, 9.92 ppm, or 9.58 ppm, these peaks are characteristic for the proton from the aldehyde groups. Several signals in the region of 3.14–4.7 ppm could be attributed to the CH or OH groups. ^13^C NMR spectroscopy (Figure 3) shows that in the CMCOx spectrum dissolved in deuterated water, there are no signals in the aldehyde groups region between 190 ppm and 200 ppm. Still, a characteristic signal of carbonyl groups in ketones appears at 217.77 ppm. There is a possibility that the hydrated form of the aldehyde groups may appear in an aqueous solution [50,51], but the characteristic signals were not found in the spectrum of CMCOx dissolved in deuterated water. In the spectrum of CMCOx dissolved in DMSO, there are signals of the aldehyde groups at 204.03 ppm, 200.29 ppm, 197.78 ppm, and 194.06 ppm.

In the CMC spectrum dissolved in deuterated water, the signal at 102.45 ppm corresponds to the C1 atom in the anhydroglucose unit. The signal from 71.315 ppm is attributed to the C7 methylene group. The carboxylic group signal appears at 178.146 ppm and can be attributed to the carboxylated ion. The signal from 59.88 ppm can be assigned to the C6 atom from the native CMC anhydroglucose unit, while the signal corresponding to the C6 atom in the CMC was shifted to 70.4 ppm [52]. In the CMCOx dissolved in the DMSO ^13^C NMR spectrum, we observe that the signals belonging to the C2 and C3 atoms from 74.96 ppm and 73.28 ppm, respectively, from the CMC spectrum, no longer occur, being an evident proof of CMC oxidation. In the ^13^C NMR spectrum of CMC dissolved in water, a slight reduction of the signal intensity from these two carbon atoms was observed. Literature mentions that when the oxidation degree is higher, these peaks’ intensity is lower. CMC oxidation determines numerous signals in the region of 80–105 ppm. The signals are attributed to carbon atoms from C-O-C groups of hemiacetals [50]. The results obtained are consistent with those reported by other studies [50,51,52].

#### 2.1.3. Oxidation Reaction Kinetic and Quantitative Determination of the Aldehyde Groups Obtained in CMCOx

The variation in time of the oxidation degree of CMC was monitored by determining the consumption of NaIO_4_ during the reaction. Figure 4 shows the results obtained.

For the oxidation degree determination, a volume of 100 mL of CMC solution of 3% concentration (*w*/*v*) was prepared, and the working protocol was described in the Section 3.4. It was found that the oxidation degree gradually increased up to 4 h when it reaches the value of 22.11%, after which it was practically stabilized, reaching the value of 22.9% after 6 h. Consequently, partial oxidation was achieved, which was our demand, because advanced oxidation can result in oxidation of carbonyl groups to carboxylic groups and, possibly, the polysaccharide chain degradation with a drastic reduction of the molecular weight. The number of moles of aldehyde groups varies, and it was between 0.71 × 10^−3^ and 5.86 × 10^−3^ moles CHO/g of CMCOx. Similar results were found and reported in the literature, where it was shown that the aldehyde groups’ number increases over time in the oxidation reaction [31,53].

The first effect of CMC oxidation is the increase in its water solubility. On the one hand, this effect is determined by the increase of the carbonyl group number, which are less involved in the hydrogen bonds formed between the polymer chains than the -OH groups and, on the other hand, by the decrease in the CMCOx molecular weight. The oxidation process is completed in about 6 h in the dark. In some papers, it is estimated that the oxidation process retardation can be caused by the hydrogen bond that is formed between the carboxyl group of one polymer chain and the hydroxyl group from the C3 atom of neighboring chains. The oxidation degree can also be adjusted by changing the ratio between the reactants [50].

#### 2.1.4. Molecular Weight Determination

The oxidation reaction can also determine the polysaccharide degradation, reducing its molecular weight. It was considered necessary to monitor the variation of this characteristic for CMCOx obtained after different oxidation times using the viscometric method. Figure 5 shows CMC’s average viscosimetric molecular weight variation with the oxidation reaction time.

The oxidation process consequences on the CMC average molecular weight were evaluated. Three samples were obtained by varying the oxidation reaction time (between 2 and 6 h). The reactants’ amounts were maintained constant (1 g of NaIO_4_ and 1 g of CMC) in each synthesis. There was a significant reduction in CMC’s molecular weight with increasing oxidation time due to polysaccharide degradation. The results obtained are consistent with the results obtained by Eremeeva and Bykova [54].

The average viscosimetric molecular weight decrease from 288.7 kDa for the initial CMC to about 83.6 kDa for CMCOx (after 6 h of oxidation), when the oxidation degree was maximum, respectively 22.9%). The reduction of the polysaccharide’s molecular weight took place by hydrolysis of the glycosidic bonds during the oxidation process, decreasing when the oxidation degree increases. It can be noted that the average viscosimetric molecular weight variation of CMCOx was consistent with the oxidation degree. After four hours of CMC oxidation, the average molecular weight value was almost constant.

### 2.2. Obtaining and Characterization of the CS Films with CMCOx

Hydrogel films based on CS with CMCOx were obtained at pH = 2.4 and a temperature of 55 °C. At a higher pH value, CS precipitates due to the reduction of its solubility. The reaction between the protonated amino groups from CS and the aldehyde groups from CMCOx determines the obtaining of imine groups or Schiff bases. Some of the unreacted amine groups form physical bonds (mainly electrostatic) with the CMCOx carboxylic groups, as mentioned in Section 2.2.3. The structure proposed by us for the obtained film is schematized in Appendix A. The cross-linking reaction is influenced by the cross-linking time or temperature, and the amount of CMCOx used (the molar ratio between the amino groups in CS and the aldehyde groups in CMC). The influence of these factors on cross-linking was studied in the paper.

Curcumin was incorporated by diffusion and alcohol evaporation after the films were dried (described in Section 3.4.9). The results of the encapsulation efficiency are given in Section 2.2.6.

#### 2.2.1. FTIR Spectroscopy of the Hydrogels Obtained

Figure 6 shows the FTIR spectra of CS, CMCOx for 6 h, and the P2 sample without curcumin in which the molar ratio -CHO/-NH_2_ was 0.375:1. The spectra were restricted only to the wavenumber range between 1800–600 cm^−1^, where the differences were expected.

In the P2 sample’s spectrum, the absorption peaks characteristic of the constituents polymers are found. The imine bond (-N=CH-), commonly called the Schiff base, is the most common covalent bond used to create CS-based hydrogels by reaction with the carbonyl group. Figure 6 shows in the hydrogel spectrum the appearance of new absorption peaks at 1645 cm^−1^, 1690 cm^−1,^ or 1742 cm^−1^. The peak from 1320 cm^−1^ from the chitosan spectrum was shifted at 1336 cm^−1^ and split into many other lower intensity peaks than the CS spectrum. The literature states that the imine group’s (CH=N) characteristic absorption band (1660–1630 cm^−1^) cannot be observed clearly, probably because it overlaps with the chitosan amide I absorption band from the chitosan spectrum [48]. The aldehyde groups’ peak characteristic from 1677 cm^−1^ in the CMCOx spectrum appears shifted in the spectrum of hydrogel films to about 1690 cm^−1^, indicating possible interactions between the amine and aldehyde groups with the possibility of Schiff bases formation. The characteristic chitosan band from 1559 cm^−1^ shifts in the spectrum of the P2 sample and several peaks of lower intensities are observed, which may indicate some intermolecular interactions of amine groups [55]. The absorption peak from 1514 cm^−1^ can be attributed to the stretching vibration C=C or C=N [44,56,57].

The absorption band from 1742 cm^−1^ indicates the electrolytic dissociation of the -CH 2COO^–N^a^+^ group and occurs, especially in polyelectrolyte complexes. The literature states that this absorption band is stronger if amino groups’ consumption intensifies [58]. Visually, the change in the hydrogel films’ yellow color is due to the imine bond formed by the two types of functional groups’ condensation [59].

#### 2.2.2. Scanning Electron Microscopy

The film’s surface morphology was highlighted by scanning electron microscopy. Figure 7 shows the SEM photos at the surface of the two obtained films. To highlight the cross-linking degree’s influence on the films’ morphology, we selected the P1 and P3 samples. The molar ratios CH=O/-NH_2_ used for the film obtaining are quite different 0.25:1 for the P1 sample and 0.5:1 for the P3 sample.

There was a slight difference in the morphology between these two samples. It was found that both are relatively compact and homogeneous, with the surface revealing a slight porosity and roughness. The porosity decreases when the CMCOx amount increases—an effect that can be explained by the films cross-linking density improvement determined by a higher number of aldehyde groups employed in reaction with the amine ones from CS.

#### 2.2.3. Amino Groups Conversion Index Determination

The free amino groups determination from CS and cross-linked CS-based hydrogel films was essential because it provides information on the amino groups’ conversion degree from CS to Schiff bases in the presence of aldehyde groups from CMCOx, necessary to optimize the parameters that could influence the cross-linking reaction. The ninhydrin test was used for this purpose. The conversion index (CI%) was analyzed for several films differing by the initial molar ratio between the carbonyl and amine groups, compared to the amino groups from free CS.

Based on the CS calibration curve, the number of moles of free amino groups in the obtained hydrogel films (N_a_) was determined. The number of moles of amino groups that react was established by the difference between the number of initial free amino groups in CS (determined based on the initial CS amount added in the cross-linking reaction) (Nb) and those determined in the films. The CI was calculated using equation (3). Several factors that can influence the CI values were studied: the molar ratio (-CHO/-NH_2_), cross-linking reaction time, and temperature. Figure 8 presents the results obtained after the amino group conversion index determination by studying the factors influencing the cross-linking reaction mentioned previously.


**The molar ratio CHO/NH_2_ influence the CI values**


The films were obtained using different molar ratios between the amino groups from CS and the aldehydes from CMCOx. The number of moles of amino groups from CS was maintained constant for all samples, respectively 1.75 × 10^−3^ moles, varying the number of moles of carbonyl groups. The amino groups’ CI was determined for all the samples, without active principle immobilized, and Figure 8a presents the obtained results. The CI values were found to increase when the CMCOx amount in the samples was higher, reaching 85% for the 2:1 molar ratio. It was observed that the CI value for the P1 sample, for example, in which the molar ratio was 0.25:1, was 42.27% and not 25% as expected theoretically. The unexpectedly high value of CI could be explained only by the existence of strong intermolecular interactions such as hydrogen bonds or by the complexation of carboxylic groups with amine groups, which could capture the -NH_2_ groups not involved in Schiff-base bonds formation leading to an erroneous conclusion about the CI value. The effect is maintained, less and less intense, up to the molar ratio CHO/NH_2_ of 0.5:1 (mol/mol).

The intermolecular interactions mentioned before decrease with the increase of the CMCOx amount by amino group consumption in the condensation reaction. Other authors also indicate the possibility of the formation of polyelectrolytic complexes at pH = 2.4 or 2.5. A research study states that CMC can form polyelectrolytic complexes at pH = 2.4 with gelatin at the gelatin: CMC ratio of 1.7: 1. The number of amino groups that could interact electrostatically with the carboxylic groups of carboxymethyl cellulose is lower at pH = 2 compared to pH = 3 [60]. Also, at pH = 2.5, polyelectrolytic complexes based on CMC and CS were prepared, and it was showed that the efficiency of obtaining them depends on the ratio between CMC and CS [61].

The possibility of obtaining polyelectrolytic complexes at pH = 2.4 was tested to elucidate whether the electrostatic interactions between the amino groups from chitosan and the carboxylic groups from CMCOx can influence the CI value mentioned above. Three films were prepared based on CS and CMC to test this hypothesis, keeping the same polymers amounts as those corresponding to CS mixtures with CMCOx obtained at different molar ratios CHO/NH_2_ (0.25:1-P1; 0.375:1-P2, and 0.5:1-P3). After drying the films, the amino groups were determined by dosing them with ninhydrin. The obtained results are presented in Appendix A, keeping the samples’ initial codes, with the mention that for the CI (%) were introduced the following notations: CI_chemical cross-linking and physical interactions_ (%), which comprise the total amino groups percentage involved in chemical bonds and physical interactions (column 3); CI_physical interactions_ (%) which express the amino groups’ percentage involved only in physical interactions (column 4); CI_chemical cross-linking and physical interactions_ − CI_physical interactions_ = CI_chemical cross-linking (Shiff base)_ (%) which expresses the of amino groups percentage involved only in the chemical reaction of Shiff base formation (column 5).

A first finding is that, indeed, it was found that in non-chemically cross-linked samples, some of the amino groups from CS did not participate in the reaction with ninhydrin, obviously because amine groups were involved in stable interactions with carboxylic groups of CMC. It is assumed that the same effect occurs when cross-linked samples are obtained, i.e., some of the amino groups of CS react with the CMCOx carbonyl groups, and others interact with the carboxylic group of CMXOx. These interactions could explain the high CI values corresponding to these samples, which do not correctly express the CI of amino groups into Schiff bases.

Thus, for the P1 sample, CI_chemical cross-linking and physical interactions_ − CI_physical interactions_ = CI_chemical cross-linking (Shiff base)_ (%) difference was 25.8%. The maximum theoretical value of the corresponding CI being 25%, according to the molar ratio of the functional groups involved in t e hP1 sample cross-linking reaction (0.25 moles of -CH=O for 1 mol of -NH_2_). Similarly, for the P2 sample, the difference was 31.08%, given that the molar ratio between these two types of functional groups was 0.375:1. For sample P3, corresponding to a molar ratio of 0.5 moles of -CHO groups for 1 mol of -NH_2_ groups, was 37.94% (compared to the maximum theoretical value of 50%).

The high amino groups consumption once with the increase of the molar ratio between the functional groups involved in the chemical cross-linking reaction and as the cross-linking degree increase could lead us to the conclusion that the interaction between the unreacted -NH_2_ groups from CS with -COOH from CMCOx could be diminished when the molar ratio increase. Therefore, at a molar ratio value higher than 0.75:1 (CHO: NH_2_), the CI value accurately reflects the reaction between amino groups from CS with the aldehyde groups from CMCOx; as a result, the Schiff bases formation. Following the data presented in Figure 8a, it was found that for the P4 sample, the CI values approached the theoretical one and continue to increase slightly with the molar ratio value for samples P5–P7, without reaching the maximum of 100%, even if the carbonyl groups content from samples in some cases was in excess. In principle, this effect is not possible; the drastic reduction of the chain fragments mobility between the obtained cross-linked network nodes and the possibility of the functional groups’ interaction in the cross-linking reaction was reduced.

It should be noted that intermolecular bonds (physical or complexed) are reversible and can be readily cleaved compared to irreversible chemical bonds [62]. Thus, the Schiff bases can stabilize the hydrogel based on CS and CMCOx, but in the case of these bonds, the possibility can exist of hydrolytic degradation in aqueous solutions. Lü et al. have shown that hydrogels’ degradation after 24 h decreases in intensity as the cross-linking degree increases because water molecules cannot enter into the cross-linked polymer network, and thus the degradation is diminished [63].


**The cross-linking time and temperature influence on the amine group CI**


To determine the influence of cross-linking time and temperature on CI (more precisely, on CI_chemical cross-linking and physical interactions_), sample P2 was selected because the CI has a medium value, and the influence of these parameters could be studied. Six hydrogel films were prepared to evaluate the cross-linking time influence in which all reaction conditions were kept constant, and only the cross-linking time was varied between 1 to 6 h. The amino groups’ CI variation as a function of the cross-linking time for the P2 sample is shown in Figure 8b.

Six other films were obtained to evaluate the cross-linking temperature influence on the CI at different cross-linking temperatures between 20 °C and 60 °C, maintaining the other conditions constant as it was described in Materials and Methods. Figure 8c shows the CI variation as a function of the cross-linking temperature for the films obtained.

Figure 8b shows that the CI increases with increasing cross-linking time, reaching a maximum value of 64% after 6 h. Given the -CH O/-NH_2_ molar ratio’s initial value, only 37.50% of the total amino groups could react with the aldehyde ones in CMCOx. It results in the involvement of 26.5% of the total amino groups in CS in intermolecular interactions, such as hydrogen or electrostatic bonds, with their number being slightly higher when the cross-linking time increases.

Of course, the increase of CI with the cross-linking time can be caused both to the increase in the number of imine-type chemical bonds between chains and the number of intermolecular interactions of physical nature. Also, the pKa value of the carboxylic groups in CMCO is 3.5: at a lower pH value, hydrogen bonds and electrolyte interactions can be formed that blocks the -NH_2_ groups, preventing them from reacting with ninhydrin. In our case, the pH of the cross-linking medium was 2.4. Their number may increase as the polymers’ contact time increases at a temperature of 55 °C. The results are consistent with those obtained when analyzing the molar ratio influence on the CI using the ninhydrin test.

Figure 8c shows that the CI value increases until a temperature of 30 °C when it reaches a maximum value, after which its value begins to decrease. The range of CI variation on this temperature range was not very high, with a maximum of 8%. Still, the values remain superior to the theoretical one, with a maximum value of 37.5%. The CI would be expected to increase continuously with the temperature (which positively influences the chemical reactions). On the other hand, the increase of this parameter has the effect of cleaving the physical or hydrogen bonds between the functional groups within polymers. As a result, ninhydrin will react with an increasing number of amino groups released from their physical interactions, and the CI value will be reduced. However, it should be noted that even if the temperature reaches 60 °C, not all physical connections were cleaved, as the apparent value of CI remained above the theoretical value of 37.5%. The previously presented influencing factors study allowed us to obtain important information, qualitatively, on the most favorable conditions for obtaining hydrogel films with the dermal application.

#### 2.2.4. Hydrogel Films Ability to Absorb Aqueous Solutions

The hydrogel films swelling behavior in physiological fluids is essential for their biomedical applications. On the one hand, this behavior determines the ability of the films to be loaded with drugs and, on the other hand, influences the transport of these drugs through the polymer network, which will influence the release kinetics of the active ingredient. The swelling degree was determined gravimetrically for P1, P2, and P3 samples at 37 °C to equilibrium in a 0.1 M phosphate buffer solution at pH = 7.4 and a 0.1 M acetate buffer solution pH = 5.5. The influence of the CHO/NH_2_ molar ratio, cross-linking time, and cross-linking temperature on the swelling degree values was studied to establish the optimal parameters for obtaining biocompatible hydrogels for controlled and sustained release of active principles with transdermal applications.


**The molar ratio CHO/NH_2_ influence on the swelling degree value**


In order to study the influence of this parameter, the P1, P2, P3 samples obtained at three -CHO/-NH_2_ molar ratios values were selected because the amino groups’ CI value was medium and were representative for this determination. Figure 9 presents the swelling degree variation in time for P1, P2, and P3 samples in two mediums of different pH values.

Figure 9 shows that the value of the swelling degree decreases when the CMCOx amount increases, regardless of the solution pH value in which the hydrogel films were immersed, an effect absolutely consistent with the CI of the amino groups’ evolution. The weakly alkaline environment causes an increase in the swelling degree due to the CMC composition’s carboxylate groups. It should be noted that the electrostatic repulsions between the polymer network chain segments are intense and are influenced by the carboxylate groups’ number from CMCOx and determines the absorption of a higher buffer solution amount.


**The influence of the cross-linking temperature on the hydrogel films swelling degree**


The influence of temperature on the cross-linking density and the swelling degree in aqueous media with different pH values was studied. As mentioned in Section 3.4.7, the films were weighed and immersed in an acetate buffer solution at pH = 5.5, respectively, in a phosphate buffer solution at pH = 7.4. The swelling degree was determined gravimetrically at well-established time intervals up to equilibrium. The P2 sample was considered of reference, and it was selected for this study (the P2 sample was chosen for the study of the cross-linking temperature influence on amino groups CI value). Four hydrogel films were prepared using different cross-linking temperatures between 30 °C and 60 °C. As mentioned in the Section 3.4, all other parameters were maintained at constant values. The four hydrogel films’ swelling degree kinetics in buffer solutions were studied. The kinetic curves obtained are presented in Figure 10.

The analysis of Figure 10a allows us to ascertain that the films can absorb water at pH = 5.5, and the swelling degree values depend on the cross-linking temperature. The swelling was instantaneous when the obtained samples were immersed in aqueous solutions, and the polymers’ strong hydrophilic character determined it. At the same time, the presence of amine groups that were protonated in a weak acid medium leads to electrostatic repulsions between the ammonium cations formed, the effect being the distance of the chain segments between the cross-linking nodes and, consequently, the diffusion of an increased water amount into the polymer network. The swelling degree increases rapidly in the first hour and then slower in the time interval range of 1–3 h with a tendency to reach equilibrium after 8 h. The swelling degree depends on the cross-linking temperature at which the hydrogels are obtained. In agreement with the CI evolution, the cross-linking temperature increase after 30 °C determines the decrease of CI values and a higher swelling degree value (Figure 10). Because there are strong physical interactions between the CS chains (responsible for the amino groups’ partial blocking that can no longer react with ninhydrin), it could be concluded that these physical interactions act as an additional cross-linking. These interactions decrease in intensity as the cross-linking temperature increase (see Section 2.2.3), and the swelling degree values in the aqueous solutions increases. For the samples prepared at different cross-linking temperatures, the same swelling degree evolution in time was recorded in the films immersed in weakly basic aqueous solution at pH = 7.4 (Figure 10b). Still, the swelling degree values obtained at this pH were superior to those obtained at pH = 5.5.

At temperature values above 40 °C, the weakly alkaline medium’s swelling degree values are higher than in the acidic medium. However, at pH = 7.4, the -NH_2_ groups’ protonation from CS is no longer possible. The possible explanation of this behavior could be that the hydrogels obtained contain -COOH groups from CMCOx. In the alkaline medium, the electrostatic repulsions between the formed -COO- (carboxylate) groups become very strong and determine the polymer network relaxation, expressed by an intensified absorption of the aqueous solution; therefore, the swelling degree values increase [23]. At higher temperatures, many -COOH groups are released from their interactions with the -NH_2_ groups, becoming carboxylated groups, and the electrostatic repulsions between the chain segments of the network were intensified.

Based on these results, it can be concluded that the CS/CMCOx hydrogels are pH-sensitive. Therefore they can be included in the category of smart hydrogels.


**The influence of the cross-linking time on the hydrogel films swelling degree**


It was previously stated that the cross-linking reaction time causes higher amino groups’ CI values, suggesting that the cross-linking degree increase even if significant intermolecular interactions such as hydrogen bonds and electrostatic interactions occur at the same time. Therefore, the cross-linking time influence on the swelling degree values in buffer solutions at pH = 7.4 and pH = 5.5 of the P2 sample (with the molar ratio 1:0.375) was studied. Six hydrogel films were obtained using different cross-linking times (between 1 and 6 h). The kinetic study of the swelling process in time in buffer solutions at pH = 5.5 and pH = 7.4 is presented in Figure 11.

The first finding was that the swelling was faster in the first 30 min, followed by a slower increase in swelling degree value, and finally, there was a tendency to stabilize it after about 3 h. The cross-linking reaction duration, obviously accompanied by the network density increase, reduces the swelling degree values in aqueous buffer solutions. Such swelling degree evolution with the cross-linking time was expected given that the CI of the amino groups into Schiff bases increase, and a higher aldehyde groups’ number from CMCOx were involved in the reaction with the amine groups from CS, causing the formation of an increasingly dense polymer network. The hydrogels’ pH-sensitive nature was highlighted again. It is observed that, regardless of the cross-linking time, the swelling degree values were higher in the weak alkaline medium (pH = 7.4). The explanation was given in the previous paragraph. It is verified that if the cross-linking time increase, reduces the swelling degree maximum value, reached after about 8 h.

#### 2.2.5. Encapsulation Efficiency

Curcumin was incorporated into the samples according to the protocol described in Section 3.4.9. The results obtained for the curcumin encapsulation efficiency in the analyzed samples are presented in Figure 12.

It was found that the curcumin encapsulation efficiency decreases as the CMCOx amount in the films increases, consequently as the cross-linking degree increases. This effect was determined because the polymer network meshes became smaller when the cross-linking density increased, and the curcumin amount that could diffuse into the cross-linked polymeric matrix was lower.

#### 2.2.6. *In Vitro* Release Kinetics of Curcumin from Hydrogel Films

The curcumin release kinetics from the obtained films were studied on a Franz cell using chicken skin as a membrane. The method has been described in the Section 3.4. Figure 13 shows the curcumin release kinetic curves in 0.1 M acetate buffer solution at pH = 5.5 containing 1% Tween 80 and in a 0.1 M phosphate buffer solution containing 1% Tween 80 at pH = 7.4 for P1C, P3C, P5C. These samples were selected to highlight very clearly the influence of the cross-linking degree, determined by the molar ratio -CHO/NH_2_, being consistent with the conversion index of the amino groups, on the process of curcumin releasing from the films. Table 1 shows the release efficiency and permeability values of immobilized curcumin through the membrane in the Franz cell’s receptor compartment. The exponential factor n calculated from the Ritger–Peppas [64] equation is also presented in Table 1.

Figure 13a,c shows the kinetic curves of curcumin release in the receptor compartment after approximately 30 h at pH = 5.5 and after 48 h at pH = 7.4. Table 1 shows that the total curcumin released efficiency (the curcumin release efficiency from receptor compartment cumulated with the release efficiency of curcumin calculated based on the curcumin amount retained within skin membrane) was higher at pH = 5.5 than at pH = 7.4. The total release efficiency at pH = 5.5 was 63.5% compared with release efficiency of 26% at pH = 7.4 for films with a low cross-linking density (P1 sample). The release efficiency increases when the cross-linking degree decreases, being affected by the polymer network meshes size, which was higher when the cross-linking degree was lower. The amount of curcumin retained in the skin membrane (based on which the release efficiency within the skin was calculated) was lower at pH = 7.4 than at pH = 5.5 (Table 1). The pKa value of CS varies between 6.17 and 6.51 [65,66]. At a pH lower than the pKa value, the free amino groups could be protonated and interact with different skin proteins leading to an increased curcumin absorption efficiency. Regardless of the pH used, the skin retention efficiency of the curcumin released from the hydrogel films increases when the cross-linking degree decreases. It can be observed that the curcumin release efficiency in the receptor compartment at pH = 7.4 has a higher value compared to the curcumin release efficiency at pH = 5.5 (Figure 13), but the retention of curcumin released from hydrogel films at pH = 7.4 in the skin membrane was lower than at pH = 5.5 (Table 1). 

In the weakly alkaline medium, curcumin was found in the phenolate form and became more soluble; therefore, its release efficiency in the Franz cell receptor compartment was intensified. The curcumin released from films through the skin membrane in the receptor compartment at pH = 7.4 depends on the carboxylate groups found in the polymer matrix that lead to an increase of the swelling degree, which intensifies the active principle diffusion. At this pH, the free amino groups are not protonated, and the hydrogel film in which curcumin was included does not interact with the skin proteins.

Curcumin-loaded films were used to study in vitro transdermal permeability through the Franz diffusion cell.

The curcumin release kinetics were also evaluated by determining the permeability of curcumin released through the skin membrane in time, expressed as μg curcumin/cm^2^. Regarding the curcumin permeability through the skin, it was found that the results obtained are comparable to those obtained at the curcumin release efficiency determination. Thus, for the encapsulated curcumin, the total permeability (the curcumin permeability from receptor compartment cumulated with the permeability of curcumin retained in the skin membrane) was higher at pH = 5.5 compared to pH = 7.4 (Table 1) due to the protonated amino groups that could interact with proteins found in the skin membrane and, therefore, the retention of curcumin released from hydrogels into the skin was improved.

The permeability variation curve s in time through the skin membrane in the receptor compartment is presented in Figure 13b,d for P1C, P3C, P5C samples, and the results are consistent with those obtained when the release efficiency was determined. Thus, a better permeability of curcumin released from hydrogels through the skin in the receptor compartment of the Franz diffusion cell at pH = 7.4 compared to pH = 5.5 was observed, this effect being determined by the slightly increased solubility of curcumin (which partially passes into phenolate form) in the alkaline medium at pH = 7.4. The curves have a typical appearance for polymer/drug systems from which the drug was released in a controlled and sustained manner, with a “burst effect” in the first 300 min. After this period, the curcumin release was slower, with a tendency to equilibrium until approximately 2000 min. The curcumin release depends on the film’s cross-linking degree because the active principle diffusion through the hydrogel becomes more and more slowly as the polymer network’s cross-linking degree increases.

The exponential factor n value from the Ritger Peppas equation calculated based on the curcumin release curves shows that the diffusion mechanism was practically Fickian for most of the samples used to study the release kinetics. For the curcumin release from the P1C sample, the exponential factor n value was 0.63; thus, the diffusion mechanism was of the non-Fickian type and disturbed by certain factors that have not been studied. Also, at pH = 7.4, the exponential factor n value for curcumin release from the P5C sample was 0.31, indicating a Fickian type diffusion mechanism, but probably disturbed by interactions between curcumin and the constituents polymers functional groups from hydrogels. It can be observed from the permeability and efficiency curves that the curcumin release from the P5C sample did not reach an equilibrium after 48 h. There is a possibility that these interactions between curcumin and the polymer matrix determine the active principle diffusion disruption.

Agrawal et al. showed that the curcumin permeability in a based hydrogel based on Carbopol 940F through mouse skin was 0.67 ± 0.01 μg/cm^2^/h and the free curcumin permeability in aqueous solutions is 0.46 ± 0.02 μg/cm^2^/h [67]. Ternullo et al. also obtained a curcumin permeability of 1.5 μg/cm^2^/h on human skin from a CS hydrogel with immobilized curcumin [68].

In our case, the total curcumin permeability after its diffusion from the film into the chicken skin at pH = 5.5 was between 2.27 μg/cm^2^/h and 3.3 μg/cm^2^/h, and at pH = 7.4, it was between 1.33 μg/cm^2^/h and 2.21 μg/cm^2^/h. The CS bioadhesive properties could determine this improved curcumin permeability from the films through the chicken skin, and the CMCOx contributes to the increase of the film’s hydrophilic character due to the free carboxylic groups that could interact at pH = 5.5 with skin proteins that contains lysine residues.

#### 2.2.7. Antioxidant Activity

The antioxidant activity expressed by IC50 was determined for P1C, P3C, P5C films, for CS, CMCOx, for P5 films without curcumin immobilized, for free curcumin, and as well as for three mixtures obtained using the identical amounts of CMCOx and curcumin as in P1C, P3C, and P5C samples. Ascorbic acid was used as a standard. The curcumin mixtures with CMC were dissolved in 50 mL of ethanol/water solution, were coded with M1, M3, and M5, and different curcumin concentrations solution expressed in μmoles/mL were used to determine the inhibition percent of the free radicals from DPPH and based on these results to evaluate the antioxidant activity expressed by IC50 for the mixtures mentioned before. The antioxidant activity evaluation of the immobilized curcumin within hydrogel films allows the study of the influence of the constituents polymers and the molar ratio between polymers on this active principle feature. In principle, when the IC50 value is lower, the antioxidant activity is higher. As mentioned in Section 3.4.10, the IC50 value was determined based on the graphical representation between the inhibition percentage versus concentration expressed in μmoles/mL. The results obtained are presented in Table 2 and Appendix A.

It was found that the IC50 values for free curcumin and curcumin included in the P1C film are close, but the IC50 value decreases when the CMCOx amount in the films increases. The curcumin-loaded hydrogel film’s antioxidant activity increases as the hydrogel film’s cross-linking degree increases. In order to determine whether CS or CMCOx influences the curcumin antioxidant activity, the IC50 value was determined for CMCOx, CS, P5 sample without curcumin and for curcumin/CMCO mixtures coded with M1, M3, M5. The P5 hydrogel film influence on the antioxidant activity was evaluated and was used the same film amounts to determine the inhibition percentage of DPPH free radicals as in the case of curcumin-loaded P5C films inhibition percentage determination.

As previously mentioned, the curcumin encapsulation efficiency in films decreases with the cross-linking degree increasing. Therefore, to obtain the same curcumin concentration in the ethanol solution used to determine the DPPH free radicals inhibition percentage for P1C, P3C, and P5C hydrogel films, the film amount used for this determination, was higher as the films cross-linking degree was improved. The obtained results showed that CMCOx did not have antioxidant activity. CS had antioxidant activity, but the IC50 value was much higher than that of curcumin. It was also found that the IC50 value was significantly higher for the P5 film compared to that obtained for CS. However, the CS film’s concentration used to determine the inhibition percentage was higher than the free CS concentration used. There is a possibility that the CS antioxidant activity decreases as the free amino groups are fewer and fewer as they react with the aldehyde groups from CMCOx. The amino group’s CI in Schiff bases in hydrogel films with a molar ratio of 1:1 was 76.5%, which means that only 23.5% of the total amino groups are free.

The IC50 values for curcumin from the M1 or M3 mixtures were close, but the IC50 value increased for sample M5 compared to the other two samples. The M1, M3, and M5 samples’ antioxidant activities were higher than that of free curcumin. We notice that the antioxidant activity decreased with the increase of the CMCOx amount in the curcumin/CMCO mixture.

The free curcumin could interact with the aldehyde groups from CMCOx, and new compounds could be formed. The recorded UV spectra show two peaks that are not characteristic of curcumin (425 nm) at wavelengths of λ = 292 nm and λ = 360 nm. In a future study, we will try to elucidate this aspect. For the P5C film, the IC50 value was 0.039 μmoles/ml, and the antioxidant activity was higher than that of free curcumin. However, the IC50 value increases, being 0.092 μmoles/mL for the M5 mixture, in which the same curcumin and CMCOx amounts were used as in P5C hydrogel film. In order to determine whether the IC50 value obtained of 0.039 μmol could be influenced by the CS antioxidant activity, the inhibition percentages obtained from the analysis of each amount of P5 film without curcumin were summed with the inhibition percentage obtained for each curcumin concentration from the M5 mixture. Then, based on the curve obtained from the graph I% vs. the concentration of curcumin (μmoles/mL), IC50 was determined. The IC50 value obtained was 0.035 μmoles/mL, being very close to the IC50 value obtained for curcumin-loaded in the P5C film, explaining the curcumin-loaded films’ antioxidant activity increase. The obtained results show that CS influences the curcumin IC50 values and determines curcumin-loaded films with improved antioxidant activity. The obtained CS films were cross-linked with aldehyde groups from CMCOx and dried before curcumin encapsulation, and the possibility of interactions between curcumin and CMCOx was very low. Thus, the wavelength where curcumin included in the film had a maximum absorption does not change compared to free curcumin’s characteristic wavelength (425 nm). Therefore, the CS films influence the curcumin antioxidant activity, and the IC50 value decreases.

The obtained results allowed the optimization of the cross-linking reaction between CS with CMCOx to obtain hydrogel films with biomedical applications. An optimal film must be flexible, mechanically stable, and have a high capacity to include and controlled release active principles. A high cross-linking degree causes incorporation of a small amount of active principle and a slow diffusion of it from the polymer matrix in quantities that may not reach the therapeutic concentration but with high mechanical stability. The optimal parameters for obtaining the film must be identified to ensure good mechanical stability and a high capacity for inclusion and controlled release of the active ingredient. Considering the results presented above, the optimal parameters established for obtaining the films used for dermal applications were: the cross-linking temperature of 55 °C, the cross-linking time of 2.5 h, and the molar ratio -CH O/NH_2_ that range between 0.25:1 and 0.5:1 (0.375:1 is preferable).

## 3. Materials and Methods

### 3.1. Materials

Medium molecular weight chitosan (CS) with a degree of deacetylation of 75% (M = 190–310 kDa and viscosity of 200–800 cPs), carboxymethyl cellulose (CMC) with a degree of substitution of 70%, powder curcumin (extracted from Curcuma Longa), DPPH (1,1-diphenyl-2-picrylhydrazyl), Tween 80, ninhydrin, sodium meta periodate were purchase from Sigma Aldrich, potassium iodide, sodium thiosulfate, ethyl alcohol, acetic acid, disodium phosphate, monosodium phosphate, sodium chloride were purchase from Chemical Company.

### 3.2. Preparation of Oxidized Carboxymethylcellulose

The method was adapted from A. Kulikowska [69] with some modifications. Briefly, 3 g of CMC was dissolved in 100 mL of distilled water at 80 °C, and the solution thus prepared was added into a 250 mL flask and allowed to cool at 30 °C. A stoichiometrically calculated amount of 3 g of NaIO_4_ was dissolved in bi-distilled water (20 mL) at room temperature and then added dropwise over the CMC solution at 30 °C. The reaction takes place in the dark, under stirring (500 rpm) for 2, 4, and 6 h, respectively. The reaction product was precipitated in ethanol (at 4 °C), and the precipitate was filtered, washed three times with ethanol, and dried at room temperature.

### 3.3. Obtaining Hydrogel Films Based on CS and CMCOx

Three grams of CS were dispersed in 200 mL of an acetic acid solution at pH = 2.4, and the solution was maintained under stirring at room temperature for 24 h. The solution at pH = 2.4 was prepared using 10 mM NaCl and glacial acetic acid to adjust the pH. According to Caneret al. [70], acetic acid is the best solvent for obtaining CS-based hydrogel films with optimal mechanical and barrier properties. Then by centrifugation, it was obtained a clear CS solution. The undissolved CS resulting from the centrifugation was dried at 100 °C, weighed, and the solution concentration was adjusted to 1.5%. Different molar ratios between the amino groups from CS and the aldehyde groups from CMCOx were used, maintaining the CS amount constant and varying the CMCOx amount. The number of moles/g chitosan was determined by calculations knowing that the structural unit’s molecular weight was 171.5 g/mol; this value was calculated knowing that the chitosan deacetylation degree was 75%, and in each structural unit, there is only one amino group. The number of moles of aldehyde/g was calculated using titration with Na_2_S_2_O_3_, as described in Section 3.4.3 from Materials and Methods. 

The molar ratios of -CHO/-NH_2_ used were: 0.25:1, 0.375:1, 0.5:1, 1:1, 1.5:1, 2:1 (mol/mol). In the second step, 20 mL of the previously prepared 1.5% CS solution was heated to 55 °C, and then, under stirring, the required amount of CMCOx dissolved in 10 mL solution of pH = 2.4 was added dropwise. The solution thus prepared was stirred for 2.5 h. 0.3 g glycerin (which ensures a weight ratio of 1 g glycerin for 100 g solution of polymers mixture) was added to obtain a flexible, fragility-free hydrogel that can be applied to the skin, ensuring good contact with it. The polymer solution was poured into 9 cm diameter Petri dishes and dried at room temperature. After drying, the films are detached from the Petri dishes and stored in the refrigerator at 4 °C until further characterizations. Table 3 shows the experimental program for obtaining hydrogel films.

### 3.4. Characterisation Methods

#### 3.4.1. FTIR Spectroscopy

FTIR spectra were obtained for CMCOx, and films based on CS cross-linked with CMCOx without curcumin were recorded using the KBr pellet method. The spectra were recorded on a Bruker Vertex FTIR spectrophotometer (Billerica, MA, USA) over a frequency range of 4.000–400 cm^−1^ at 4 cm^−1^ resolution-32 scans (discussions were made in Section 2.1.1 for CMC or CMCOx, and in Section 2.2.1 for films based on CS cross-linked with CMCOx).

#### 3.4.2. ^1^H NMR and ^13^C NMR Spectra of CMC and CMCOx

Nuclear magnetic resonance spectroscopy (Bruker NEO 1-400, Billerica, MA, USA) was employed to investigate the product’s molecular structure before and after oxidation. The samples were prepared by dissolving CMC in deuterated water at 80 °C, and the CMCOx was dissolved in deuterated water or DMSO, after which they were analyzed by ^1^H NMR and ^13^C NMR technique.

#### 3.4.3. Oxidation Reaction Kinetics and Quantitative Determination of Aldehyde Groups Obtained in CMCOx


**Aldehyde groups determination**


The working protocol was adapted with some modifications after Teotia [31] and McSweeny et al. [71]. The aldehyde group content in CMCOx was determined indirectly by dosing the residual sodium periodate in a reaction mixture by iodometric titration. Briefly, 1 mL of CMCOx solution was added over 1 mL of 20% KI solution mixed with 1 mL of 37% HCL. I_2_ was titrated with 0.05 N Na_2_S_2_O_3_ until the solution color turns light yellow. Afterward, to visualize I_2_, 1 mL of starch solution with a concentration of 1% was added, and the solution’s color was turned blue. It was titrated until the color becomes transparent, and I_2_ no longer existed in the reaction. Lange previously described the iodometric titration reactions in 1961 [72].

Based on the reactions involved, the amount of sodium periodate that did not react in the oxidation reaction was determined. The NaIO_4_ amount that reacts was calculated as the difference from the initial amount of NaIO_4_ added in the oxidation reaction. The number of moles of aldehyde groups could be calculated, considering that 1 mole of NaIO_4_ reacted was required to obtain 2 moles of aldehyde groups in CMCOx.

Oxidation kinetics studies

Three grams of CMC was dissolved at 80 °C in 100 mL of bi-distilled water. After the solution temperature decreases at 30 °C, 2 g of NaIO_4_ dissolved in 20 mL of water was added dropwise over the polymer solution, under stirring. The reaction occurs in a glass flask, on a water bath, in the dark at 30 °C, under stirring to prevent air and light influence on the oxidation reaction. After every hour for up to 6 h, 1 mL of sample was taken from the reaction medium and titrated with Na_2_S_2_O_3_. The degree of oxidation was calculated with Equation (1).
(1)$$\mathrm{OD}\%=\frac{Q_{R}}{Q_{T}}\;\times 100$$
where Q_R_ is the NaIO_4_ amount that reacted, and Q_T_ is the total amount of NaIO_4_ introduced into the reaction.

#### 3.4.4. Molecular WEIGHT determination of CMC and CMCOx by the Viscometric Method

The CMC and CMCOx intrinsic viscosities were determined by working at different polymer concentrations (0.1%, 0.2%, 0.3%; 0.35%, 0.4% and 0.5% *w/v*) in NaCl solution in a weight ratio polymer: NaCl = 1:1 (for example for a polymer solution preparation with 0.2% concentration, 0.2 g polymer were dissolved in 100 mL solution of 0.2% NaCl concentration), using an Ubellohde viscometer (K = 0.00999) at 25 °C. The average viscosimetric molecular weight (M_η_) was calculated using the Kuhn–Mark–Houwink Equation (2):
[η] = K × M_η_^a^(2)
where K = 1.23 × 10^−3^ and a = 0.91 (for both initial CMC and CMCOx) [54], [η] represent the intrinsic viscosity, and M_η_ is the average molecular weight determined by the viscosimetric method.

[η]—was deduced from the equation of the graphical representation ηred as a function of the solution concentration.

#### 3.4.5. Scanning Electron Microscopy

Hydrogel films were characterized by scanning electron microscopy (SEM) to determine their surface morphology. They were dried, metalized with gold using a spray deposition device, and analyzed using a HITACHI SU 1510 electron microscope. (Hitachi SU-1510, Hitachi Company, Chiyoda City, Tokyo, Japan).

#### 3.4.6. The Amino Groups’ CI% Determination into Shiff Bases in Hydrogel Films

The CS used (average molecular weight) has a degree of deacetylation of 75%, and the molecular weight of the structural unit is 171.5 g/mol. A CS stock solution of 0.1% (*g*/*v*) concentration was initially prepared to plot the calibration curve using the ninhydrin test by dissolving the CS amount in acetic acid 0.1%. Different volumes were taken to prepare six different CS concentrations between 0.1 and 0.4 mg/mL and were brought to sign using acetate buffer solution pH = 5.6, 0.1 M in volumetric flasks of 10 mL. The ninhydrin test protocol used to plot the calibration curve was as follows: in the test tubes to 1 mL of CS solution of different concentrations prepared before was added 2 mL of 2% ninhydrin solution in ethanol. Then the reaction mixture was heated at 95 °C for 30 min, and the color of the solutions became dark blue. After the solutions were cooled, 8 mL of ethanol/distilled water solution (1:1 volume ratio) was added. The spectrophotometer’s calibration was done with a blank prepared, as we mentioned before but without chitosan. Instead of chitosan, 1 mL of acetate buffer of pH = 5.6 was used. The solutions’ absorbances were recorded with a UV spectrophotometer at a wavelength of 570 nm. The concentrations expressed first in mg/mL were converted to molar concentrations (number of moles of amino groups’/mL). The CS calibration curve was plotted using the ninhydrin assay, and it is presented in Appendix A (absorbance as a function of the number of moles of amino groups from the CS). The CS calibration curve has the equation y = 0.056x.

It was considered necessary to determine the number of free amino groups from the obtained films using the ninhydrine test. The working method was similar to the one presented above used to plot the chitosan calibration curve. The small film amounts previously weighted were added in the test tubes, and 1 mL of buffer acetate 0.1 M, pH = 5.6 with 2 mL of 2% ninhydrine solution was added, then heated at 95 °C for 30 min. After the solution was cooled, 8 mL of ethanol/distilled water solution (1:1 volume ratio) was added, and the absorbance was measured at 570 nm. Based on the chitosan calibration curve previously plotted, the free amino groups within hydrogel films were determined. The total number of amino groups involved in the formation of Schiff bases, but also in strong physical interactions with the carboxylic groups in CMCOx, can be calculated by the difference between the total number of moles of free amino groups initially added in the reaction (calculated based on the initial chitosan amount). The free amino groups within the hydrogel were determined using the ninhydrin test. The conversion index was calculated with the relation (3) and expressed:(3)$$\mathrm{CI}\%=\frac{N_{b}-N_{a}}{N_{b}}\times 100$$
where N_b_—number of moles of free amino groups before cross-linking; N_a_—number of moles of free amino groups after cross-linking. N_b_−N_a_—represent the bonded amino groups within hydrogel film.

The influence of the cross-linking temperature, cross-linking time, and the molar ratio on CI value was studied. The CI of the amino group determination was performed in triplicate. The standard deviation was within ±3%.

#### 3.4.7. Hydrogel Films Ability to Absorb Aqueous Solutions

The obtained films have hydrogel character, so it was considered useful to determine their ability to retain water—usually quantified by the degree of swelling (Qt,%). This feature is significant because it causes the more or less intense diffusion of the active ingredient from the hydrogel matrix.

For the obtained films, Qt was determined gravimetrically. The buffer solutions simulating physiological fluids used were phosphate buffer at pH = 7.4 and an acetate buffer simulating skin pH at pH = 5.5.

The films were dried to constant weight, and a quantity of precisely weighed film (M_dry_) was immersed in a 5 mL solution of different pH values at 37 °C. At various intervals, the sample was removed from the liquid medium, and its surface was buffered with filter paper to remove the excess water. The mass of the swelling films (M_swelling films_) was determined by weighing. The solution absorbed by the films (M_solution_) represents the difference between the swollen film’s weight (M_swelling films_) and the weight of the dry film (M_dry_). After weighing, the samples were reintroduced into the solution, the operation being repeated after well-established time intervals until equilibrium was reached. The swelling degree was expressed as the ratio between the amount of solution retained in the films at each time interval measured and the amount of completely dry film (relation (4)). The influence of the cross-linking temperature, cross-linking time, and the molar report on swelling degree value was studied. The swelling degree was performed in triplicate. The standard deviation was within ±5%.
(4)$$Q\%=\frac{M_{\mathrm{solution}}}{M_{\mathrm{dry}}}\times 100$$

#### 3.4.8. Encapsulation Efficiency

The whole film was used for curcumin encapsulation by diffusion and alcohol evaporation method. The procedure was as follows: a quantity of 20 mg of curcumin was dissolved in 15 mL of ethanol, and the obtained films (see the Methods section) were immersed in the curcumin solution and maintained for 24 h at 45 °C for ethanol evaporation. After 24 h, the dry films were placed for 30 s in 25 mL of phosphate buffer solution at pH = 7.4 to remove the curcumin found on the film’s surface that was not immobilized. In order to determine the curcumin quantity that was not immobilized, 20 mL of ethanol was added to the containers in which curcumin encapsulation took place. The curcumin amount was determined spectrophotometrically at λ = 425 nm based on the curcumin calibration curves in ethanol or phosphate buffer solution previously plotted. The difference between the initial curcumin amount used for encapsulation and the curcumin amount determined spectrophotometrically, it was determined the encapsulated curcumin amount in the hydrogel films. The encapsulation efficiency was determined with the relation (5):Ef% = (curcumin amount encapsulated)/(initial curcumin amount) × 100 (5)
-initial curcumin amount = 20 mg

#### 3.4.9. *In Vitro* Release Kinetics of Curcumin from Hydrogel Films

The in vitro release kinetics of curcumin from the obtained hydrogel films were studied in two different pH media: 0.1 M acetate buffer solution at pH = 5.5 (similar with skin pH) and 0.1 M phosphate buffer solution at pH = 7.4 (similar with blood pH), at 37 °C, until equilibrium.

In order to simulate the dermal application, curcumin release was studied using a Franz cell (2 cm diameter, 15 mL receptor volume), the cell compartments being separated by a chicken skin membrane. The skin was degreased with 96% ethyl alcohol using sterile dressings, and only skin without defects and open-pore was used for the experiment. After preparation, the skin was stored in 10% glycerin solution for 24 h before use. The membrane thus prepared was fixed between the donor and the receiving compartment. The films with immobilized curcumin (size 2.5 × 2.5) were placed in the Frantz cell’s donor compartment on the skin membrane’s surface. Acetate buffer solution at pH = 5.5 or phosphate buffer solution at pH = 7.4 with 1% Tween 80 were used as a release medium. 1 mL solution of pH = 5.5 or pH = 7.4 was placed in the donor container. The curcumin release kinetics took place in the dark (the Franz cell was covered with aluminum folium), the temperature in the receiving compartment was maintained at 37 °C and the stirring at 150 rpm. 0.5 mL of the receptor compartment medium were taken after different time intervals for quantitative measurement of released curcumin and replaced with a fresh medium. The release profiles were obtained by measuring the curcumin concentration using a nanodrop UV-vis spectrophotometer at 425 nm. The permeability of curcumin (P) from films was also calculated through the skin membrane expressed as μg curcumin/cm^2^ with the relation (6), and the release efficiency (%) was calculated with relation (7):(6)$$P=\frac{M_{r}+M_{s}}{S},\;µg/{\mathrm{cm}}^{2}$$
(7)$$R_{\mathrm{Ef}}\%=\frac{M_{r}+M_{s}}{M_{i}}\;\times 100$$
where M_r_ represents the curcumin amount released in the receptor; M_s_ represents the curcumin amount found in the skin membrane; S represents the surface of the film (12.56 cm^2^); M_i_ represents the total curcumin amount immobilized in the film disc used for release.

The curcumin calibration curves shown in Appendix A have the following equations: y = 0.0167x (calibration curve in ethanol); y = 0.0079x (calibration curve in phosphate buffer at pH = 7.4), respectively, y = 0.0104x (calibration curve in acetate buffer pH = 5.5). All the determinations were performed in triplicate. The standard deviation was within ±5%.

#### 3.4.10. Antioxidant Activity

With some modifications, the work method was described before by Choi et al. [73]. The curcumin stock solution was prepared by dissolving 5 mg of curcumin in 50 mL of ethanol. Several dilutions were made to test the antioxidant activity, and the final concentrations of curcumin solutions were between 10 and 50 µg/mL. 2 mL of each solution concentration was added in the test tubes, over which 2 mL of 0.1 mM DPPH solution (in ethanol) was added. The samples prepared were vortexed for 20–30 s and maintained in the dark at a temperature of 37 °C for 1 h. The samples’ absorbance was measured after 60 min using a UV spectrophotometer at a wavelength of 517 nm. Ascorbic acid was used as a standard. The absorbance values were converted into antioxidant activity percentage (free radicals inhibition percentage in DPPH-I%) using the following relation (8):(8)$$I\%=100-\left[\left(A_{s}-A_{b}\right)\times \frac{100}{A_{c}}\right]$$

IC50 (expressed as µmoles/ml) was calculated from the graphical representation I% vs. concentration and represented the sample’s concentration that can capture 50% of the free radicals in DPPH. Ethanol was used to calibrate the spectrophotometer—the A_s_ represents the absorbance value of different concentrations of solutions. As a blank (A_b_), it was used a solution prepared from 2 mL of ethanol and 2 mL of curcumin solutions of different concentrations was used (the blank’s absorbance was measured for each concentration). The control solution was prepared using 2 mL of DPPH solution and 2 mL of ethanol. P1C, P3C, and P5C films were selected to determine the immobilized curcumin antioxidant activity. Different film amounts containing between 10 and 50 μg of curcumin were used, and the percentage of free radical inhibition percentage (I%) was determined as described above. It was also determined the antioxidant activity for CS solution, hydrogel films without curcumin, and curcumin mixed with CMCOx dissolved in an ethanol/water mixture. It was prepared mixtures of curcumin with CMCO by adding identical amounts of curcumin and CMCO as in the films P1C, P3C, P5C, and the samples were marked with M1, M3, and M5. The antioxidant activity of M1, M3, M5 samples was also determined. All the determinations were performed in triplicate. The standard deviation was within ±5%.

## 4. Conclusions

CMC was oxidized with sodium meta-periodate, and the aldehyde group formation was demonstrated by FTIR and NMR spectroscopy. The polysaccharide oxidation degree increases in time, its maximum value being 22.9%. CMC was degraded during the oxidation reaction; its molecular mass decrease when the oxidation time increases oxidation, up to 1186 g/mol after 6 h of oxidation. Hydrogel films were obtained using different molar ratios between the amino groups from CS and the aldehyde groups from CMCOx. FTIR spectroscopy demonstrated the Schiff bases’ formation, and analysis of the films’ surfaces by SEM showed that the hydrogels’ porosity/rugosity decreases with the increasing of the CMCOx amount in their composition. The amino groups’ CI into Schiff bases depended on several influencing factors such as the CMCOx concentration in the film composition, the intermolecular interactions between the constituent polysaccharides, the cross-linking time, and the cross-linking temperature. It was found that some of the amino groups from CS are involved in strong intermolecular interactions with carboxylic groups of CMCOx, their reaction with ninhydrin was restricted, and the values of the amino groups’ CI were distorted. The swelling degree determined in two different pH environments (pH = 5.5 and pH = 7.4) increases when the CMCOx amount decreases in the film and when the amino groups’ CI value decreases. The cross-linking temperature and the cross-linking time for the films prepared significantly influence the swelling degree, obtaining a maximum value at the temperature of 60 °C and a short cross-linking time (one hour). The CI values decrease with the temperature and cross-linking time increasing. Higher values of the swelling degree at pH = 7.4 compared to those at pH = 5.5 were obtained due to the presence in the film of free carboxylic groups from CMCOx, which causes strong electrostatic repulsions and the absorption of a higher amount of aqueous solutions.

The thermal stability of the films depends on the cross-linking degree. Thermogravimetric analysis showed that the maximum weight loss is lower for the samples with a higher crosslinking degree (P3). Curcumin was included in the hydrogel films by diffusion method and evaporation of alcohol, and it was observed that the encapsulation efficiency decreases with increasing the cross-linking degree. The kinetics of curcumin release from the film and permeability were studied in buffer solutions of pH = 5.5 and pH = 7.4 through a chicken skin membrane, using a Franz diffusion cell. The results showed that the curcumin release from films intensified at pH = 7.4. Still, at pH = 5.5, there is a higher absorption of curcumin amount in the skin membrane, making the total release efficiency higher at pH = 5.5. Curcumin-loaded in the hydrogel films retains its antioxidant activity, and CS contributes to its increase. Obtaining CS-based hydrogel films cross-linked with aldehyde groups from CMCOx can be optimized for various applications such as biomedical ones.

## Figures and Tables

**Figure 1 molecules-26-02185-f001:**
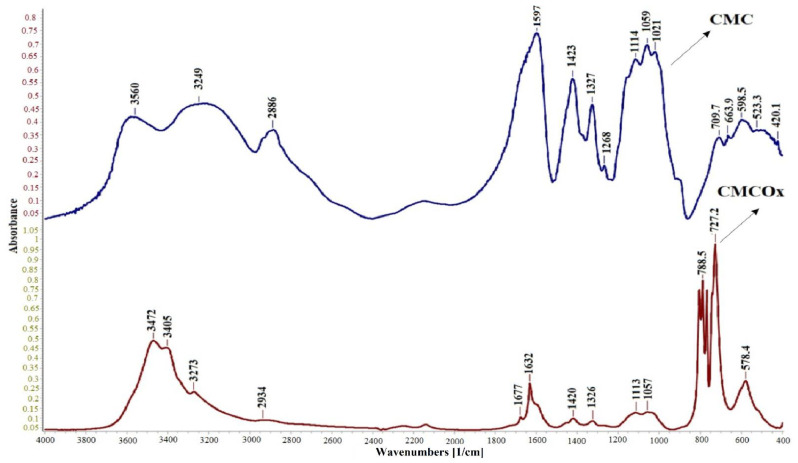
Fourier-transform infrared (FTIR) spectra for carboxymethyl cellulose (CMC) and oxidized carboxymethyl cellulose (CMCOx)- the time used for the oxidation was 6 h.

**Figure 2 molecules-26-02185-f002:**
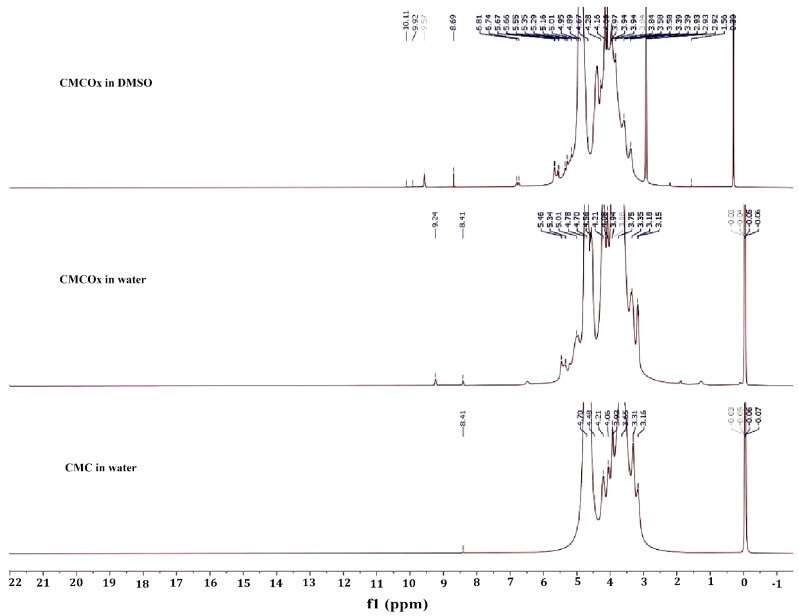
Proton nuclear magnetic resonance (^1^H NMR) spectra for CMC dissolved in water, CMCOx dissolved in water, and CMCOx dissolved in DMSO.

**Figure 3 molecules-26-02185-f003:**
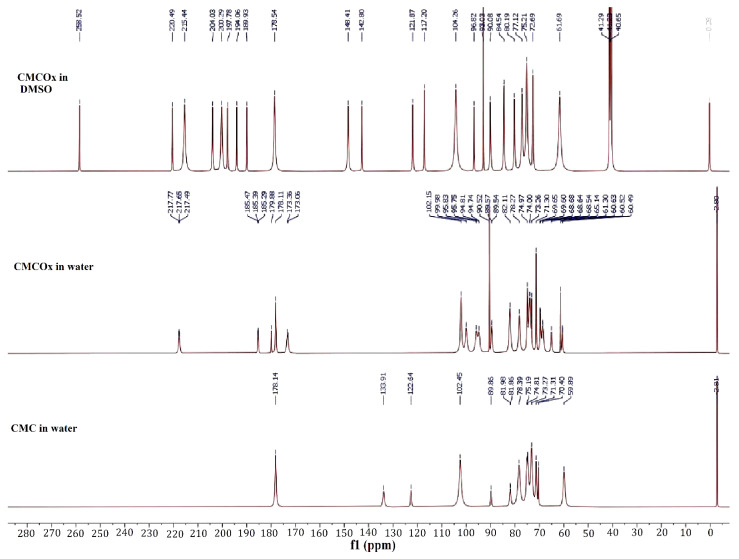
Carbon-13 nuclear magnetic resonance (^13^C NMR) spectra for CMC dissolved in water, CMCOx dissolved in water, and CMCOx dissolved in DMSO.

**Figure 4 molecules-26-02185-f004:**
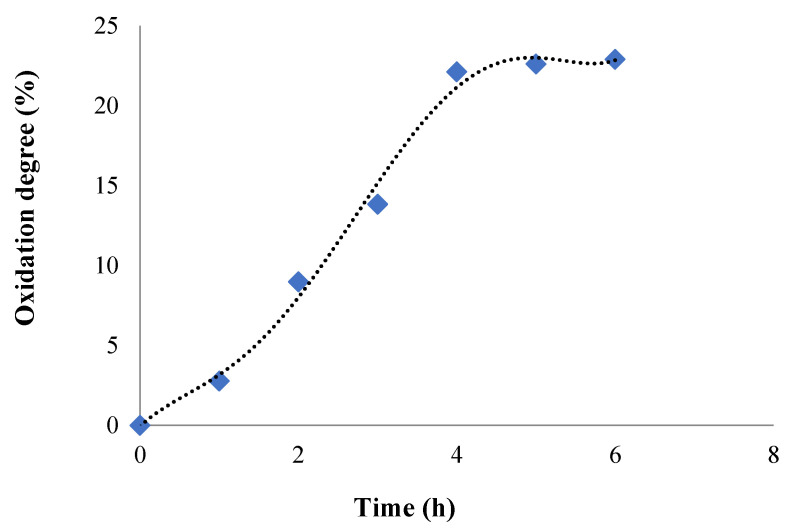
Variation in time of the oxidation degree of CMC.

**Figure 5 molecules-26-02185-f005:**
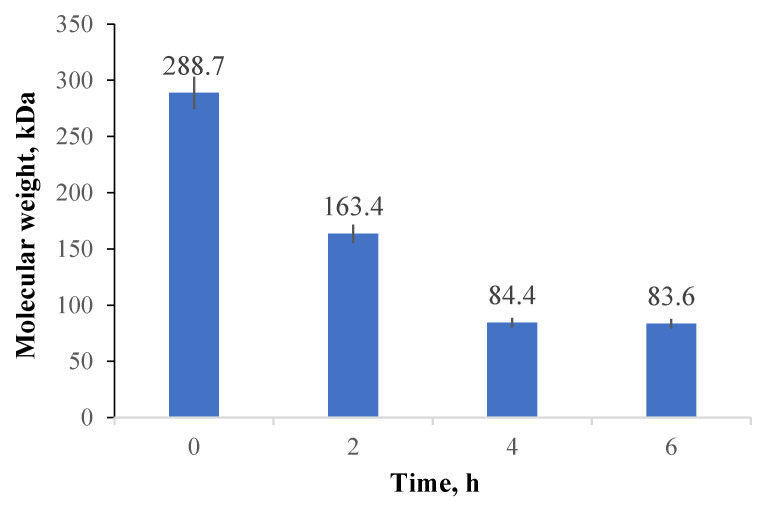
Variation of the average CMC molecular weight with the oxidation process duration determined by the viscosimetric method.

**Figure 6 molecules-26-02185-f006:**
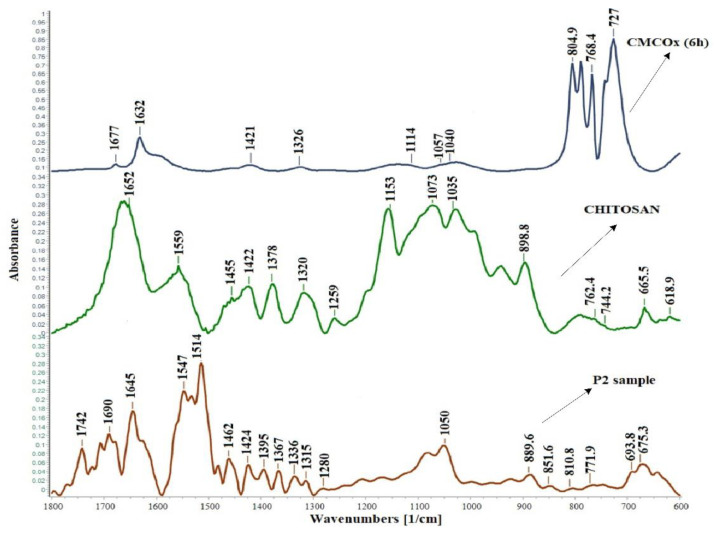
FTIR spectra of CS, CMCOx, and P2 films (with a molar ratio NH_2_/CHO = 1:0.375 mol/mol).

**Figure 7 molecules-26-02185-f007:**
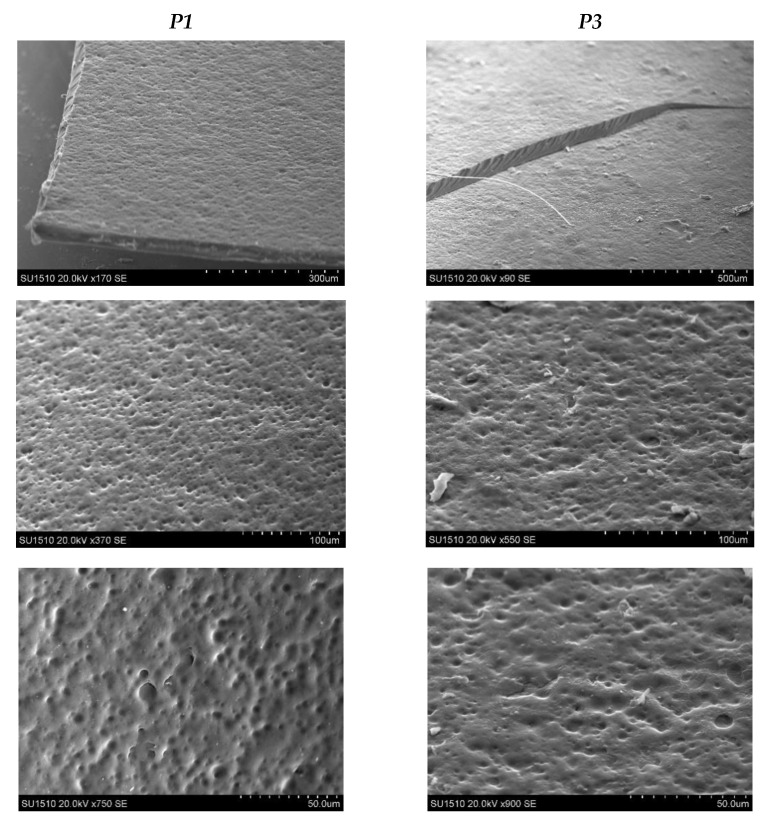
Scanning electron microscopy photographs on the P1 and P3 films surface.

**Figure 8 molecules-26-02185-f008:**
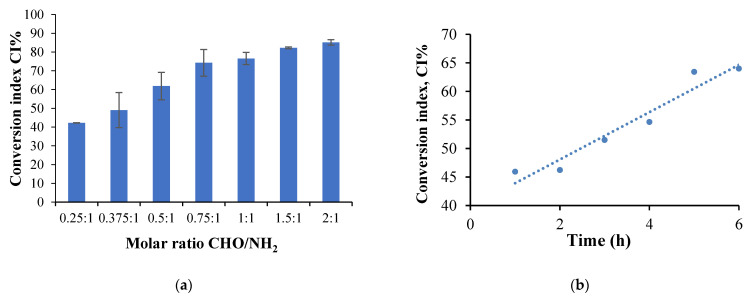
The variation of the amino groups’ conversion index values by studying the cross-linking reaction influencing factors such as (**a**) the molar ratios influence (t_reaction_ = 2.5 h; T_reaction_ = 55 °C, pH = 2.4—it was determined for all the samples (**b**) variation of CI values as a function of cross-linking time for P2 sample (T _reaction_ = 55 °C, molar ratio CHO: NH_2_ = 0.375:1, pH = 2.4); (**c**) variation of CI as a function of cross-linking temperature for P2 sample (T_reaction_ = 55 °C, molar ratio CHO: NH_2_ = 0.375:1, pH = 2.4).

**Figure 9 molecules-26-02185-f009:**
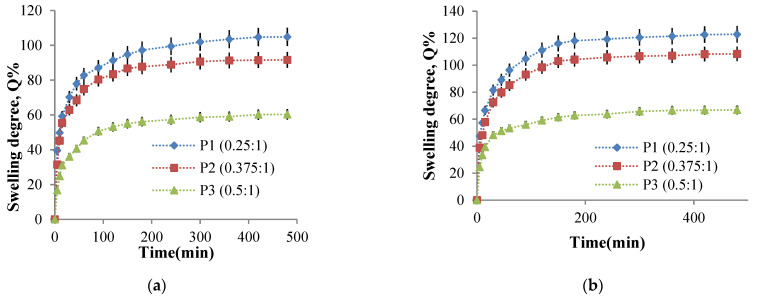
The variation in time of the swelling degree for P1, P2, and P3 samples using different molar ratios in two solutions of different pH: (**a**) pH = 5.5 and (**b**) pH = 7.4.

**Figure 10 molecules-26-02185-f010:**
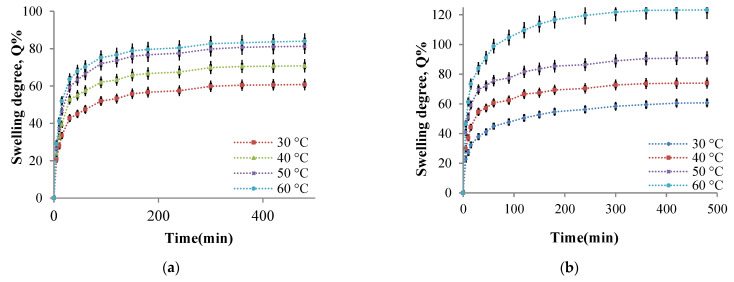
The variation in time of the swelling degree for the P2 sample prepared at different cross-linking temperatures in acetate buffer solution, pH = 5.5 (**a**) and in phosphate buffer solution at pH = 7.4 (**b**).

**Figure 11 molecules-26-02185-f011:**
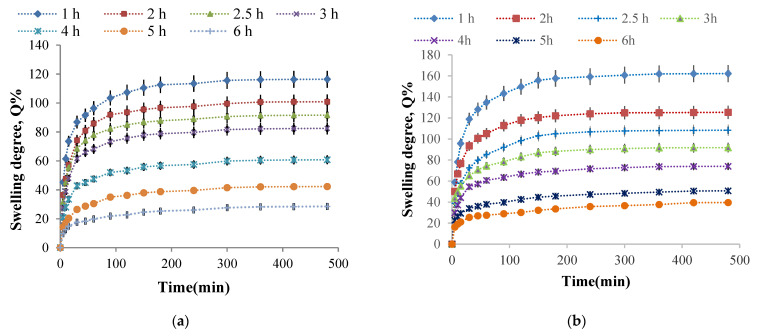
Variation in time of the swelling degree for P2 sample obtained at different cross-linking times in different pH solutions: (**a**) pH =5.5 and (**b**) pH = 7.4.

**Figure 12 molecules-26-02185-f012:**
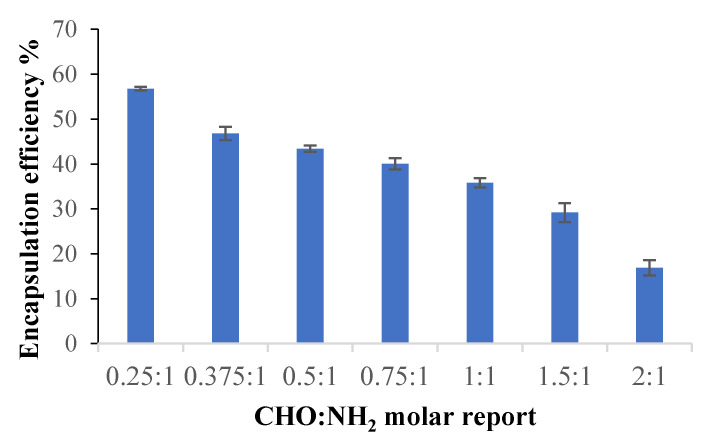
Curcumin encapsulation efficiency in CS/CMCOx films obtained at different CHO: NH_2_ molar ratios.

**Figure 13 molecules-26-02185-f013:**
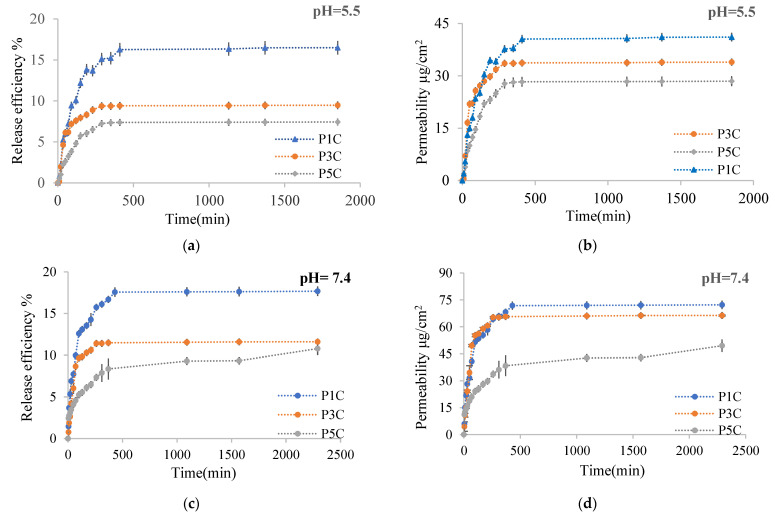
The curcumin release kinetics in time in acetate buffer solution at pH = 5.5 expressed as release efficiency (**a**) or in terms of permeability-μg/cm^2^ (**b**) and phosphate buffer solution at pH = 7.4 expressed as release efficiency (**c**) or in terms of permeability-μg/cm^2^ (**d**), for samples P1C, P3C, and P5C.

**Table 1 molecules-26-02185-t001:** The release efficiency and permeability of curcumin-loaded in hydrogel films through the skin membrane.

Sample	Ratio	Efabs% on Skin Membrane, %		Ef% in the Receptor, %		Total Ef, %		P (µg/cm^2^) in the Receptor (after 48 h)		P (µg/cm^2^) in Skin Membrane, 48 h		Total P (µg/cm^2^), 48 h		n		R2	
		pH 5.5	pH 7.4	pH 5.5	pH 7.4	pH 5.5	pH 7.4	pH 5.5	pH 7.4	pH 5.5	pH 7.4	pH 5.5	pH 7.4	pH 5.5	pH 7.4	pH 5.5	pH 7.4
P1C	0.25:1	47	8.6	16.5	18	64	26	41.1	71	117	35.1	158	106	0.63	0.43	0.92	0.97
P3C	0.5:1	31.6	3.6	9.5	12	41	15	33.9	66.3	113	20.4	147	87	0.48	0.53	0.81	0.91
P5C	1:01	23.3	3.2	7.4	11	31	14	28.5	49.5	90	14.9	118	64	0.53	0.31	0.97	0.99

**Table 2 molecules-26-02185-t002:** Antioxidant activity expressed by IC50 for the analyzed samples.

Sample	IC50, µmoles/mL
Ascorbic acid	0.031 ± 0.00053
Curcumin	0.051 ± 0.00033
P1C	0.054 ± 0.00039
P3C	0.046 ± 0.00039
P5C	0.039 ± 0.00012
M1	0.082 ± 0.0032
M3	0.081 ± 0.00041
M5	0.092 ± 0.00846
P5 + M5	0.035 ± 0.00012
CS	0.4 ± 0.0103
P5 (without curcumin)	10.02 ± 0.28

**Table 3 molecules-26-02185-t003:** The experimental program used to obtain covalently cross-linked hydrogel films based on CS and CMCOx.

Samples Code *	The Molar Ratio (-CH=O/-NH_2_)	Moles of Aldehyde Groups from CMCOx (×10^3^)
P1	0.25:1	0.4375
P2	0.375:1	0.656
P3	0.5:1	0.875
P4	0.75:1	1.3125
P5	1:1	1.75
P6	1.5:1	2.625
P7	2:1	3.5

* The number of moles of -NH_2_ groups in the CS of 1.75 × 10^−3^ moles was kept constant. The CS solution volume was 20 mL, and the CMCOx solution volume was 10 mL. 0.3 g of glycerin was added in each synthesis to avoid the obtaining of fragile and brittle films. The films containing curcumin and containing the same molar ratios were coded with P1C, P2C, P3C, P4C, P5C, P6C, and P7C.

## Data Availability

Not available.

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
