# Peer review of "Hydrogel Films Based on Chitosan and Oxidized Carboxymethylcellulose Optimized for the Controlled Release of Curcumin with Applications in Treating Dermatological Conditions"

_molecules, 2021, doi:10.3390/molecules26082185_

Round 1

Reviewer 1 Report

This manuscript is devoted to chitosan films cross-linked with aldehyde groups of carboxymethyl cellulose for application as drug delivery system in dermatology. The obtained results may represent interest for specialists engaged in polymer science and medicine. However this manuscript can be accepted for publication after a major revision. The concrete comments are the following:

1. Page 5. Lines 209-211. The resolution of the FTIR spectra of CMC should considerably be improved. I do not see any band around 1642 cm-1 that belongs to carbonyl of carboxylic groups.  Identification of functional groups of CMC might be performed attracting the literature survey. Authors claim that the appearance of a new peak at 1677 cm-1 is attributed to aldehyde groups. In my mind this part of spectra should be enlarged in order to see how the oxidation times (2, 4 and 6 h) influence on the substitution degree of CMC. In fact the intensity of this band could be increased upon oxidation time.

2. Pages 8 and 25. Molecular weight determination. According to section 3.4.4. (Page 25, Lines 930-939) the viscosity average molecular weights (Mh) of CMC and CMOC were determined by viscometric method using M-K-H equation. To find the intrinsic viscosities of CMC and CMOC the reduced viscosities were measured in distilled water at various concentrations. However both CMC and CMCO containing the carboxylate ions belong to anionic polyelectrolytes and in pure water they will show the polyelectrolyte anomaly. In this case to avoid the polyelectrolyte anomaly some low-molecular-weight salts (usually 1:1) should be added. It is strange that the M-K-H equation is used for both CMC and CMCO. Please correct this section carefully and give an appropriate reference to M-K-H equation. GPC will be effective tool to determine the molecular weights of CMC and CMCO. Moreover, the molecular weights should be rounded up or down and indicated in kDA.

3. Page 9. Why the FTIR spectrum of oxidized CMC in Fig. 2 and Fig. 7 is different? Moreover, FTIR spectrum in Fig.7 is in good resolution and the peak at 1618 cm-1 is intensive.

4. Page 9. Lines 332-344. From this paragraph it is not seen which band belongs to -N=CH- bond (1514 cm(-1) ?). For identification of -N=CH- bond the Raman spectroscopy is more preferential than FTIR.

5. Pages 12-13. Lines 385-408. Interpretation of conversion index (CI, %) results is unclear. In Table 2 column 3 reflects the amount of crosslinked amine groups of chitosan (taken from Fig. 9) while column 3 is related to amine groups of chitosan involved into interpolyelectrolyte complexation with CMC.  In such case one can expect that the difference between column 2 and column 3 (that is column 4) is free amine groups of chitosan not reacted with aldehyde groups of CMCO. I suggest to show schematically both crosslinking and formation of interpolyelectrolyte complexes with participation of chitosan and CMCO.

6. It seems that the films fabricated from chitosan and CMCO belong to grafted macromolecules where CMCO also play the role of crosslinker due to condensation of aldehyde groups with amine groups of chitosan. Such hydrogel films will definitely show amphoteric character and isoelectric point where amine groups of chitosan and carboxylic groups of CMC or CMCO will form intraionic salt bridges or intrapolyelectrolyte complexes. It is recommended to study the pH-dependent swelling-deswelling of modified chitosan.

7. Page 13. Lines 474-476. It should be noted that at low pH the dissociation of carboxylic groups of CMC or CMOC will be suppressed while amine groups of chitosan will be ionized and the whole macromolecular chain will be charged positively.  Please indicate the pH value used for Figs 10 and 11.

8. The manuscript contains too much Figures. Most of them might be presented in Supplementary section.

Author Response

The authors would like to sincerely thank the reviewer for the detailed analysis of the submitted manuscript. All the changes made are written with blue color in the manuscript. Our responses to the comments are given in the attached word document.

Reviewer 2 Report

The manuscript titled "Hydrogel films based on chitosan and oxidized carboxymethyl cellulose optimized to include and controlled release active principles with potential applications in treating dermatological conditions" could have been an interesting study. After all, use of biopolymer based hydrogels for drug delivery remains an active area of research. However, after careful reading of the manuscript, I must regrettably say that either the study is conducted haphazardly and/ or at least presented haphazardly it has become too long with uneven flow which makes it very hard to follow. The manuscript needs complete reformatting and writing before being considered for publication. Below is the list including, but not limited to, comments, questions and suggestions.

1) First of all, the manuscript is too long for a research article making it very hard to follow. The authors should keep focus and move some of the results into the supplementary information section.

2) Abstract, Authors use FT-IR (Lines: 19) while they use FTIR (Lines: 25). Please keep the consistency. Also, The abstract should contain more specifics. For e.g.  "Higher release efficiency was found in the slightly alkaline medium" (Lines: 27-28). Give the figures. how much versus how much

3) Introduction need to be more focused and succinct.  For e.g. Lines: 43-46; "In general, women were more commonly affected by skin diseases than men, except for skin cancer, which was more commonly diagnosed in men. In general, skin diseases were more common in northern European countries (Germany, the Netherlands, and Sweden) than in southern Europe (Italy and Spain)". There is no connection between the this information and the study the authors have conducted. The authors are not going t make any geographical and gender comparison.  The authors should focus on why chitosan based hydrogels more advantageous than other polysaccharide/biopolymer based hydrogels drug delivery use. 

4) Results and discussion section and subsections. Section 2.1. "Preparation of CMCOx". There is no need for the the this information. The authors do not present any new information here based on the result. Selective oxidation at C2-C3 of ring with aldehyde formation by the use of periodate is well known. Instead here authors could combine section 2.3, 2.4, 2.5 and 2.6.

5) Lines:  165-166; "The oxidation reaction did not use or form toxic compounds, and it took place in the dark to avoid the advanced oxidation, in double-distilled water, at pH=6.5 and a temperature of 30 degree C". How do the authors certain about the statement that no toxic compounds were formed? Not supported by experimental data or the suitable reference.

6) Lines: 174-196; Section 2.2. "Obtaining CS films with CMCOx". In this section authors haven't presented any result and discussion. In fact the information provided here (Table 1) should be presented under materials and method section

7) Lines: 193-196; Table 1. How was the desired ratio of aldehyde to amino groups achieved?

8) Lines: 198-222; Section 2.3 "FT-IR spectroscopy of CMC and CMCOx". First, the figure 2 is illegible and should redraw the figure. Please check the legend of abscissa. It is wavenumber not the wave length. And the values should be presented in reverse order. While discussing the peaks please provide the references to when assigning the peaks to functional groups.

9) Lines: 209-214; "The appearance of a new 209 absorption peak of a lower intensity from 1677 cm-1 attributed to unsaturated aldehyde groups is a clear indication that partial oxidation of CMC has occurred. The absorption peak from 1720 cm-1, specific to the aldehyde group, did not occur because there is the possibility that aldehydes groups were found in the hydrated forms in CMCOx, and hemiacetals could mask it and appear shifted at 1677 cm-1". Actually distinct peaks attributed to aldehyde groups appear in oxidized cellulose based substrates prepared by periodate oxidation method (Mou, K.; Li, J.; Wang, Y.; Cha, R.; Jiang, X. 2,3-Dialdehyde nanofibrillated cellulose as a potential material for the treatment of MRSA infection. J. Mater. Chem. B 2017, 5, 7876–7884). Why is not the intensity of the peak at 1677 cm-1( "attributed to unsaturated aldehyde groups is a clear indication that partial oxidation of CMC has occurred") changed among CMCOx samples? One should expect increased intensity of this peak as the oxidation time increased?

10) Lines: 218-220; "The absorption bands from 218 1423 cm-1 to 1327 cm-1 attributed to the carboxyl groups in the CMC spectrum are maintained in the CMC spectrum but appear slightly shifted, indicating structural changes following the oxidation reaction". What do the authors mean by this statement? They mean spectra of CMCOxs as compared to the spectrum of CMC?

11) Lines: 232-249; NMR peak assignment should be backed by suitable references. No references were provided.

12) Lines: 299-299; Figure 6, Round off the molecular weight.

13) Lines: 316-344; Why the FTIR spectrum of only the P2 samples out of so many samples presented in Table 1? Please see see comment #8 regarding the figure and references to back of the functional groups attributed to particular IR peak.

14) Lines: 349-359; Section 2.8 "Scanning Electron Microscopy". The author present the SEM images of P1 and P3 samples again why? Why not of P2 since in the preceding section (FTIR), authors show P2 sample? Really hard to follow and make the logical connection.

15) In my opinion most of the results under  2.9 section can can go under supplementary information with author shortening this section by summarize important points.

16) Lines: 382-383; Figure 9. The authors should give sample coding (e.g. P1, P2, P3) in the parenthesis just for clarity.

17) Lines: 481-503; Section 2.9.3. Why again P2 sample is selected?

18) Lines: 505-515: Why here P1, P2 and P3 samples? not all other samples?

19) Lines: 584; Figure 13, Please keep the legends in ascending order rather than haphazardly (e.g. 1h, 2h, 2.5h...)

20) Lines: 602-636; Section 2.10.3 "The molar ratio CHO/NH2 influence on the swelling degree value". Why just P1, P2 and P3 not other samples? Again very hard to make connection and follow the logic. Even if there is logic/reason to select these particular samples, this section should go before 2.10.1 and 2.10.2. cause in those sections authors chose only the P2 sample. 

21) Lines: 626-628; "Considering the results presented above, the optimal parameters established for obtaining the films used for dermal applications were: the cross-linking temperature of 55 degree C, the cross-linking time of 2.5 hours, and the molar ratio -CHO/NH2 that range between 0.25:1 628 and 0.5:1". Is not clear to the reader. How?

22) Lines: 638 -672; Please round off the temperature values. Also provide the DTG curves which help visualize the decomposition temperature. Same as in the above comments, why P2 and P3 samples here? Why not P1?

23)Lines: 679 -679; Figure 16. The figure should be flipped with the encapsulation efficiency being presented in the ordinate.

24) Lines: 696-700; Figure 17. Why these particular samples for curcumin release study? Why not P2 here?

25) why is not the control (plain CS hydrogel) sample not included for curcumin encapsulation and release study as a control. Should be provided. Otherwise we don't know whether CS hydrogels crosslinked CMCOx is justified over neat CS hydrogels for this kind of applications? 

26) Lines: 878-880; "In the second step, 20 ml of the previously prepared 1.5% CS solution was 878 heated to 55°C, and then, under stirring, the required amount of CMCOx dissolved in 10 ml solution of pH=2.16 was added dropwise" CMOx dissolved what 10 ml solution of pH: 2.16? How is PH adjusted?

27) Lines: 888-850; How may coadded scans were acquired to generate a FTIR spectrum?

28) Were the samples conditioned for FTIR and TGA? if so how?

29) Lines: 881-882; "obtain a flexible, fragility-free hydrogel film with good mechanical properties". What do the authors mean by this statement? What was the measure of good mechanical properties?

30) Lines: 930-934; please provide a suitable reference.

31) Lines: 941-945; What is the voltage?

32) Lines: 1000-1005; atmospheric pressure (100 mL/min). What do the authors mean by this? The authors mean oxygen flow rate during TGA? or it was done under inert gas flow?

33) Lines: 1132-1133; Supplementary information should be referred in the main text of the manuscript. Supplementary information should be referred in the text. 

Author Response

The authors would like to sincerely thank the reviewer for the detailed analysis of the submitted manuscript. All the changes made are written with blue color in the manuscript. Our responses to the comments are given in the attached word document. Please see the attachment.

Round 2

Reviewer 1 Report

If acceptable the title of the manuscript might be formulated as follows: 

“Hydrogel films of chitosan and oxidized carboxymethylcellulose optimized to controlled release of curcumin in treating dermatological conditions”

Figure 6. Please assign the intensive band at 1514 cm-1 in FTIR spectra.

Figure 8. Please correct “Molar report” to “Molar ratio”. Please track numbers on ordinate axis of Figure 8a and 8c.   

Paragraph 3.4.4.  Molecular weight determination of CMC and CMCOx by the viscometric method. Please correct the M-K-H equation as [η] = K×Mηa (where Mη is viscosity average molecular weight). Please indicate the NaCl concentration in which the intrinsic viscosity was determined.  Instead of Mw should be Mη

I suggest authors completely remove the paragraph 2.2.5. The thermogravimetric analysis and Figure S3 from Supplementary materials. In my mind these data are not so informative for readers.
